# Nurses’ Perceptions Regarding Their Professional Commitment and Development during the COVID-19 Pandemic

**DOI:** 10.3390/healthcare11192659

**Published:** 2023-09-30

**Authors:** Pedro Ángel Caro-Alonso, Beatriz Rodríguez-Martín, Julián Rodríguez-Almagro, Carlos Chimpén-López, Cristina Romero-Blanco, Ignacio Casado-Naranjo, Alberto Bermejo-Cantarero, Fidel López-Espuela

**Affiliations:** 1Health Care Service of Castilla-La Mancha, GAI Talavera de la Reina, 45600 Talavera de la Reina, Spain; pedroa.caro@uclm.es; 2Faculty of Health Sciences, University of Castilla-La Mancha, 45600 Talavera de la Reina, Spain; beatriz.rmartin@uclm.es; 3Department of Nursing, Ciudad Real, University of Castilla-La Mancha, 13001 Ciudad Real, Spain; cristina.romero@uclm.es (C.R.-B.); alberto.bermejo@uclm.es (A.B.-C.); 4Department of Medical-Surgical Therapy, Psychiatry Area, Nursing and Occupational Therapy College, University of Extremadura, 06006 Badajoz, Spain; cchimpen@unex.es; 5Cáceres University Hospital Complex, 10004 Cáceres, Spain; icasadon@gmail.com; 6Metabolic Bone Diseases Research Group (GIEMO), Nursing Department, Nursing and Occupational Therapy College, University of Extremadura, 10003 Cáceres, Spain; fidellopez@unex.es

**Keywords:** work engagement, COVID-19, nurses, nurse’s role, SARS-CoV-2, qualitative research

## Abstract

Introduction: During the pandemic, nurses have undergone a high level of professional burnout, suffering emotional exhaustion, depersonalization, and lack of personal realization. Objective: The object of this study is to understand in depth, through a phenomenological study of Giorgi, the perceptions on commitment and professional development of frontline nurses during the first and second waves of the COVID-19 pandemic. Method: Qualitative study designed and analyzed using Giorgi’s phenomenological focus. For data collection, semi-structured interviews were utilized in a theoretical sample of frontline nurses who worked in public hospitals of Extremadura and Madrid, Spain, until saturation of data. The interviews were conducted between the months of May and December 2020 following an outline of topics. The analysis was based on the phenomenological focus of Giorgi and was supported by the software Atlas-Ti 8.0. Results: A total of 14 nurses participated in this study. Two main themes emerged to explain the perceptions of the nurses: (1) the professional commitment of the nurses during the pandemic and (2) the effects of the pandemic on professional development; seven subcategories were also identified. Conclusion: The social and professional development of nurses is important. If nurses feel that they are quality professionals, this will enable them to protect their psychosocial health and increase professional commitment toward their patients in difficult situations such as pandemics. The results of this study may serve as a guide for better understanding the problems and needs of nurses as healthcare providers. This may help administrators in the generation of solutions for the establishment of a safe and reliable work environment, which will in turn promote a healthcare system that can efficiently respond to future catastrophes.

## 1. Introduction

The coronavirus (COVID-19) illness caused by the virus SARS-CoV-2 rapidly spread throughout the world beginning in 2020, becoming recognized as a pandemic by the World Health Organization (WHO) in March of that year [1].

The National Epidemiological Surveillance Network (RENAVE) reported a total of 40,961 COVID-19 cases among healthcare professionals during the first pandemic wave in Spain [2].

Healthcare professionals exposed to or positive for COVID-19 have been affected in their health [3], seeing as nurses involved with frontline work presented more psychological symptoms (anxiety, fear, stress, impotence, frustration, and an increase in obsessions and obsessive behaviors) [4,5,6].

There have been many negative repercussions of the pandemic on nurses. During the first wave of the pandemic, nurses faced challenges posed by the critical situation saturating Spanish healthcare services. The lack of healthcare professionals in this situation was evidenced, leading administrators to request that healthcare professionals cancel their rest periods, that retired nurses be reincorporated, or that students in their last year of nursing be contracted [7]. Additionally, healthcare professionals have expressed fear of acquiring or propagating the illness because of increased exposure, and of having to adapt to continuous changes in action protocols [6,8]. The lack of knowledge on an approach to the new illness, the lack of personal protection equipment (PPE), and the need to adapt to new spaces due to the high number of patients admitted forced continuous changes in protocols and proceedings [5,9]. This generated in nurses the need to learn to seek and share knowledge, the adaptation of COVID-19 protocols and workplace mobility, and improvization in the use of protective equipment or spaces for the care of people with COVID-19 [9].

During the pandemic situation, nurses have experienced high professional burnout, in many cases suffering emotional exhaustion, depersonalization, and a sense of lack of professional realization [10]. Diverse social and occupational factors have affected the mental health of nurses, of which the fact of working in an environment with a high risk of illness spread has been the factor that has contributed the most to their exhaustion [10,11]. Almost one-third of nurses who worked during the first eleven months of the pandemic experienced symptoms of anxiety, and more than one-fifth experienced changes due to depression [12]. Moreover, undertaking more shifts has been associated with higher levels of exhaustion in healthcare professionals [11].

Despite experiencing difficult work shifts, lack of adequate protection, and risks to their physical and mental health, nurses showed their strong commitment to patient care [13]. Furthermore, nurses were mediators between patients and their families, including the final life stage, where patients passed away without the company of their families, and only in the presence of healthcare professionals [14]. Despite their strong commitment, nurses highlight the dehumanization due to the measures of isolation or personal protective equipment [15].

Regarding the protective factors during the pandemic, certain organizational, social, personal, and psychological factors may have exerted a protective effect on the mental health of frontline nurses like peer support and teamwork; valuing positive reinforcement and humanizing the situation; having effective communication, both formally and socially; and having positive, safe, and supportive learning environments [16,17]. Other important factors to minimize negative effects were resilience factor (perceived as the ability to bounce back), self-perceived social support provided by loved ones, and self-perceived social support provided by colleagues [18,19].

As a result, the objective of this study was to understand in depth, through a phenomenological study of Giorgi, the perceptions on commitment and professional development of frontline nurses caring for COVID-19 patients during the first two waves of the pandemic. Key information to understand the problems and needs of nurses in managing these types of catastrophic situations.

## 2. Method

### 2.1. Study Design

This was a qualitative study, designed and analyzed using Giorgi’s descriptive phenomenological focus. We chose this approach because of its ability to describe the meanings of the analyzed phenomenon from the nurses’ experience [20,21]. Through this approach, we sought to describe the perceptions of frontline nurses caring for COVID-19 patients during the first and second waves of the pandemic about their commitment and professional development through phenomenological analysis using their own words [20]. Additionally, study reporting was guided by recommendations from the Consolidated Criteria for Reporting Qualitative Research (COREQ) guideline [22].

### 2.2. Participants and Data Collection

To gather the data, semi-structured interviews were conducted with a theoretical sample of nurses caring for hospitalized COVID-19 patients during the first and second pandemic waves in Spain. The nurses worked in hospitals belonging to the public healthcare system in the autonomous communities of Extremadura and Madrid. These two autonomous communities were selected for their high prevalence of cases in Spain and for the possibility of accessing the sample since during part of the data collection the towns and cities were in lockdown, making travel difficult [23]. We used the nursing management of the centers to be able to access potentially interested nurses who met the inclusion criteria. Semi-structured interviews were utilized because of their capacity to describe the phenomenon from the words of the participants [24].

The interviews were conducted between May and December 2020 to include experiences from the nurses about the first and second waves of the COVID-19 pandemic. Sample collection continued until the saturation of data, at which point conducting more interviews, stopped providing new analytical concepts [25,26]. It was verified at the moment when all analytical concepts were saturated: 14 interviews (11 women and 3 men) (Table 1: main characteristics of participants). The interviews had a median duration of 66 min (range of 36–116 min).

For the selection of participants, the following criteria were considered. Inclusion criteria: (1) nurses who actively worked during the first and second waves of COVID-19; and (2) nurses who worked in hospital or intensive care units in healthcare systems in Madrid or Extremadura.

Exclusion criteria: (1) nurses who had not been actively working during that period (time off work due to illness or unpaid leave); and (2) nurses who had worked less than 2 months with COVID-19 patients, time that the research team considered necessary to be able to express the experience of the study phenomenon (Table 1).

The interviews were conducted in a private and comfortable setting, previously agreed upon with the participant. The interviewer (male, FL-E) was an expert in qualitative research and had extensive experience in the semi-structured interview technique. The interviewer conducted the interviews by using an outline of topics (Table 2) that could appear in an open manner during the interview, and which became more refined along the length of the study [21]. All interviews were audio recorded after obtaining participants’ consent. The interviewer also took field notes during the interviews, which were also material for analysis.

Most of the interviews were carried out in person, but due to the evolution of the pandemic, 4 participants preferred to carry out the interview via videoconference. No significant differences were found between the interviews conducted in person compared to those conducted by videoconference [27]. 

### 2.3. Analysis of Data

Once transcribed and made anonymous, the interviews were analyzed following the steps of Giorgi’s phenomenological method. First, the interview transcription was read and re-read to identify significant information in response to the research questions and study objectives. Once data were highlighted, significant units were identified and coded according to the suggested meaning determined by the analyst researcher. Codes were all defined in a glossary, which allowed an initial interpretation. These codes were regrouped, generating themes and subthemes. Grouping was based on the shared meaning of codes and facilitated by comparing attitude and relationship linking (through network diagramming). Finally, in an interpretative exercise aiming to integrate all themes and subthemes, a central theme emerged, which encapsulates the essence of the experience [28].

Two investigators independently conducted the data analysis, reaching a consensus about the results afterward. In case of disagreement, a third investigator was accessed. During the data analysis, the software Atlas-ti 8.0 was utilized as an aid.

### 2.4. Trustworthiness

In this study, confirmability, credibility, confidence, and transferability were applied to achieve the aspects of rigor indicated by Guba for trustworthiness [29]. To guarantee confirmability and facilitate the audit, detailed information was explicitly expressed for the different phases of data gathering, analysis, and inference. Various strategies have been developed to ensure credibility such as reflexivity (with a constant reflective attitude attending to methodological crossroads decision making and interpretative issues), research triangulation (comparison and discussion between researchers during the analytical process), and return to interviewees (during the last interviews, as themes were becoming saturated, new questions were introduced in interviews to confirm the intersubjective experience interpretation).

Confidence was achieved by involving more than one investigator in the data analysis. To attempt an increase in transferability of results, we have described meticulously the participants’ characteristics that condition their experience as well as the environment where the research has been developed [29].

Ethical considerations: This study was conducted in accordance with the Declaration of Helsinki in a private place where the participants’ emotions could be expressed while safeguarding their privacy. It also followed the confidentiality standards established by the European General Data Protection Regulation (EU 2016/679). This study was approved by the Ethics Committee on Research with Human Beings of the Caceres University Hospital (Ref: CEIM20/278). All participants signed the informed consent document following a complete and adequate explanation of the study, which included the possibility of revoking the consent at any time. All interviews were recorded by audio means, transcribed literally, and made anonymous with an alphanumeric code before being analyzed. The interview recordings were supervized by the principal investigator.

## 3. Results

The participants of this study were 14 frontline nurses who cared for COVID-19 patients in hospital or intensive care units. These participants belonged to the public healthcare systems of Madrid or Extremadura during the first and second waves of the pandemic (Table 1).

The nurses faced the situation with a high level of professional commitment, perceiving the pandemic as an opportunity for professional development. Two main themes were identified by the nurses: (1) the professional commitment of nurses during the pandemic and (2) the effects of the pandemic on professional development. Additionally, 7 subcategories were extracted from the experiences of the participants. In Table 3 and Table 4 can be found a summary of themes, categories, and codes along with the main verbalizations of the participants.

### 3.1. Professional Commitment of Nurses during the Pandemic

The frontline nurses caring for patients during the first and second waves of the COVID-19 pandemic expressed an emerging theme of living through their experience with fear. They were aware of the risks they faced while caring for their patients. Despite this, another emerging theme was the nurses’ professional commitment to continue caring for their patients. They expressed they could not stay home from work while knowing there were healthcare organizations in need of nurses. Instead, they offered to take more shifts or to prolong their shifts if necessary.

According to the nurses, their strong professional commitment and knowing that their work was necessary helped them overcome moments of weakness or fear during this time. In relation to professional commitment, the nurses described many occasions of taking care of patients without having adequate personal protective equipment. This had endangered their personal integrity as well as their lives. Additionally, the nurses considered that other professionals, such as physicians, on many occasions stepped away from frontline care due to fear of the spread of illness. The nurses described physicians delegating parts of their roles away to the nurses, especially those related to direct care of COVID-19 patients.

Another theme identified by the nurses was an awareness that they were living through a historical moment. Although at times they experienced weakness or emotional lows due to the hardship of the situation, they were proud of being able to contribute their part to alleviate the healthcare conditions they were living through during the pandemic.

On the other hand, another theme emerged as nurses found themselves facing the isolated situation of their patients. The nurses found themselves taking care of their patients’ families as well, which increased the emotional connection felt by the nurses with them. The nurses described becoming their psychological support. These ties and emotional connections with the patients and their families were perceived by the nurses as care given but not always readily visible, yet necessary to alleviate the extreme solitude and loss of hope that patients were experiencing. A majority of the time, these instances took place outside of the nurses’ work hours.

### 3.2. Effects of the Pandemic on Professional Development

According to the nurses, their self-perception as professionals improved throughout the catastrophic situation and conditions surrounding frontline healthcare professionals while caring for COVID-19 patients during the first and second waves of the pandemic. This perception helped the nurses bring out the positive part of the experience, emphasizing all they had learned professionally and personally. The nurses considered that their self-concept as nurses improved, with the following emerging aspects: (a) change in the model of care and the meaning of care; (b) changes in the organization and labor structures; (c) change in the social views of nurses; and (d) improvement in professional development after the pandemic.

aChange in the model of care and the meaning of care

The first and second waves of the pandemic caused a change in the model of nursing care. On the one hand, care of symptoms of illness took precedence over care of the person. On the other hand, population health took precedence over the model of humanized care centered on the person. These shifts in the model of care were dominant during the pandemic. In this sense, the nurses expressed that they could not dedicate all the time they would have wanted to the patients. They felt that essential aspects of care were overlooked, such as individualization and humanization of care, privacy and intimacy of the patient, or attention to emotional aspects of care. This sudden change in the model of care caused a high level of frustration among the nurses, since it contradicted the essence of nursing care. Despite this, the nurses described that the fact of being overburdened by the situation influenced their acceptance of this new model of care, and their search for its positive aspects.

According to the nurses, the use of PPE also directly influenced the expressions of care. They described that wearing PPE made it difficult to show their patients expressions of closeness such as the giving of looks, touch, etc., to which the nurses were accustomed as part of patient care. The nurses felt this turned patient care more cold, distant, and depersonalized. Despite considering PPE a barrier to patient care, the nurses reported they still tried to find other ways to deliver words of relief to their patients and to provide the psychological and emotional support they needed at those times.

bChanges in the organization and labor structures

The nurses described that there were changes in the professional relationships among healthcare professionals, especially within the team of nursing professionals. The nurses perceived that nurses and nursing assistant care technicians worked as a team, and that cohesiveness of the team increased during the pandemic, as well as team solidarity. Mutual help and support as partners within the nursing team increased in a way the nurses had not previously experienced.

According to the nurses, this mutual support within the nursing team resulted in a fusion of the roles of nursing and nursing assistant care technicians as they worked together, without following the division of labor in tasks existing until the present. This way, the nurses described that when they would enter the patient’s room, they would carry out many of the tasks that had until presently been undertaken by the nursing care technicians. This helped prevent the other team members from also having to enter the room.

On a professional level, the nurses perceived that their work teammates displayed an essential role in mutual emotional support and relief, both during and after work hours. This helped them face the difficult times they had lived through. In fact, in those cases where the nurses decided to isolate themselves from their own families to prevent the possible spread of illness, or when they lived alone, their teammates at work were their only support. The nurses perceived that their work teammates understood the situation because they were also living through it themselves.

Additionally, as another protective gesture toward their families and loved ones, the nurses attempted to not share with them what they were seeing and living through in order to not scare and concern them. Likewise, the nurses described that not only were they physically isolated from their families (changing households or living in hotels), but there was also emotional isolation due to not being able to have family emotional support or speak with their family about their concerns.

Furthermore, the nurses perceived that the professional relationships with the physicians changed during this period. Physicians diminished their overall contact, as well as visit times and meetings between nurses and physicians, due to fear of illness spread. The nurses described that the physicians became more distant and ceased to work with them as a team. There was a recurring theme described by the nurses of the physicians very frequently avoiding situations of risk for illness spread. Additionally, the nurses considered that it was the nurses who had salvaged these situations. They reported having cared for their patients in their role as frontline healthcare providers. Meanwhile, in many cases, the physicians delegated care functions to the nurses outside the nurses’ scope of practice, such as the responsibility of evaluating patients.

cChange in the social view of nurses

In the Spanish media, there were pervasive images of healthcare professionals carrying out dances and songs, or video uploads to social media such as TikTok, Instagram, Facebook, or YouTube, of messages related to professional resistance and strength. These were perceived by the nurses as strategies for destressing during the situation being lived through, and as a way to protect society from harsh images of the realities being faced by healthcare professionals. The nurses considered that this social image of the profession damaged them since these strategies were somewhat misleading and did not correspond with the work and the real situation that was being lived out in the hospitals. The nurses also reported feeling that the pandemic had been a missed opportunity to make the work of nurses visible at a social level.

Additionally, the nurses considered that despite choosing their profession based on personal conviction, they still needed social and institutional support in a real, palpable way in order to keep working in a situation as difficult as what was lived through during the COVID-19 pandemic.

During the pandemic, another social image was also distributed of healthcare professionals as heroes. The nurses reported disagreeing with this image, since they considered that taking care of patients in any situation (whether a pandemic or not) forms part of their professional identity and is their responsibility. Because of this, they considered they were only carrying out their work, not an act of heroism. Additionally, the nurses expressed that at the end of the confinement in Spain (during the second wave of the pandemic), society again went back to ignoring them. There was a return to previous aggression and insults toward healthcare professionals, and at a social level, their work again became devalued. This created the situation of being considered first a hero, then a villain.

dImprovement in professional development after the pandemic

According to the nurses, the ethical challenges experienced during the pandemic and the strategies to face them contributed to their professional development. Psychological competency was improved, as well as competency in group work, and in self-directed learning. In addition, the pandemic contributed to the nurses reaffirming themselves as professionals.

D.1Improvement in psychological competencies

The nurses perceived that providing frontline care for persons with COVID-19 provided them with increased learning and experience. They felt that after this situation, they had increased strength, self-esteem, capacity for adaptation, and resilience in providing care in adverse situations.

Additionally, the pandemic contributed to the improvement of the nurses’ self-concept and self-esteem, considering their increased capacity to adapt to various work situations (reported by the nurses as increased flexibility).

D.2Competencies for group work

The nurses perceived that their ability for group work improved after the pandemic. They reported feeling an increase in connectivity, mutual support, solidarity, and cooperation among the nursing team, with which to face the situation.

D.3Independent learning and lifelong development

The nurses described that due to the pandemic, they had to improve their competencies related to self-directed learning and continuous actualization of knowledge daily. These aspects were necessary for the nurses as they adapted to the new situation since in many instances they had to leave their area of work and specialty to care for COVID-19 patients. According to the nurses, this type of learning and training was carried out in their own free time due to an absence of formative initiatives promoted by healthcare institutions during the first waves of the pandemic.

Additionally, nurses considered that they improved their competencies related to the search for scientific evidence and the diffusion of evidence among peers.

D.4Improvement of professional self-concept

Some nurses considered that the situation they lived through during the pandemic was an opportunity to improve their self-concept about their professionalism and personal growth as nurses, considering they had professionally grown as nurses.

## 4. Discussion

This study explores the professional commitment of the nurses who worked in hospitals or intensive care units during the first waves of the pandemic. This study shows evidence of the effects of the pandemic on the professional development of nurses.

The participating nurses emphasize their commitment to nursing. Nurses have played a key role during the pandemic due to caring for their patients as frontline healthcare professionals and due to serving as a main, sometimes only, source of psychological support to persons hospitalized with COVID-19, since they could not be in direct contact with their families due to isolation [30,31].

In this sense, the nurses in the hospital and intensive care units perceive that they were often the only channel of communication for providing emotional stability for the patients during the first two waves of the pandemic, considering that professional psychological intervention was absent. This contradicts findings in another study showing the need for persons with COVID-19 to receive support and psychological assessment from mental health experts (psychologists) and religious experts to offer comprehensive care [32,33].

The participating nurses consider that the situation they lived through during the pandemic helped them rediscover the value and meaning of being a nurse, showing commitment to their frontline roles with conviction and responsibility 24 h a day. This is consistent with findings supporting that the COVID-19 pandemic has contributed to improving the professional commitment of nurses [32,34].

As has been pointed out in previous studies, the nurses working in COVID-19 units emphasize the stress and anxiety resulting from working in that environment and having to carry out work of high intensity without adequate institutional support as well as a lacking work environment. In this sense, the nurses emphasized inadequate employee rights, poor planning, a shortage of staff, and protective equipment, and insufficient basic medical facilities restrictions that affected nursing care [35,36]. On the other hand, we found that a lack of nurses negatively influences the quality of care and that attention to high-quality care is only possible if the work environment does not threaten patient safety [37]. Due to this, the nursing and hospital administrators should ensure that hospital and intensive care units can rely on an adequate number of nursing professionals and adequate support systems so that clinical nurses can provide the best care in crisis situations, such as what was lived through during the COVID-19 pandemic.

The second theme emphasized by the participating nurses was the affected psychology of healthcare professionals working in COVID-19 patient units. The results of this study confirm that during the pandemic, the nurses worked under stress and fear due to the lack of proper PPE. Likewise, there was a lack of knowledge and specific learning for the care of COVID-19 patients, as well as for the correct use of PPE. This prompted the nurses to change their focus onto attempting to alleviate symptoms of illness, to the detriment of humanized care. This contrasted with their usual focus on humanized care, such as had been done before the pandemic. This was a factor that increased stress for the nurses since they could no longer care for their patients in the same way as they had been able to before this time [36,38,39]. Previous studies show that the initiative to provide care and the responsibility of making choices regarding the use of resources increases the risk of suffering damages from moral distress and anguish in the nurses caring for these specific types of patients [40,41,42].

The nurses in this study express their commitment to caring for persons with COVID-19 with conviction and dedication. Changes in organization, labor structures, and the model of care were produced because of what the nurses experienced while caring for their patients during the pandemic. Nurses assumed the delegated roles of other healthcare professionals such as nursing care technicians. Nurses even assumed evaluation roles that should have belonged to physicians, but who were not carrying out their role due to fear of the spread of illness. Instead, the physicians delegated their role to the nurses, such as the responsibility of assessing patients to prevent them from entering rooms and taking risks. In addition, nurses perceived that their professional relationships with physicians changed during the pandemic, realizing that contact, visiting times, and meetings decreased due to physicians’ fear of contagion [17,36,43]. The COVID-19 pandemic has generated challenges for healthcare around the world, becoming an opportunity to elevate the status of nursing and help nurses better understand the essence and real value of nursing [36,43]. In agreement with previous studies, the statements of participating nurses show evidence of the inconsistencies that took place during the management of PPE and of the nursing professionals during the spread of COVID-19 [32,39].

Aside from the problems and challenges, the nurses considered the pandemic as an opportunity to change the social image of nurses. Improving the professional position and understanding the essence of nursing on the nurses’ part have been some of the opportunities highlighted by the nurses during this pandemic [39]. Despite this, the nurses in this study indicate that the images of nurses distributed on social media have hurt them, as well as the social image of the nurse as a hero. Because of this, it is important for the healthcare administration to promote a social image of nurses accurate to reality, so that nurses can continue to offer quality care during the pandemic. It has been shown that support from the healthcare administration increases nurses’ motivation and interest in providing care during critical situations [36,39,40]. Results following the line of previous studies carried out in Spain, Italy, and the United States show that nurses do not want to be perceived as heroes or receive public celebration; they just want to be seen as “nurses”, since the hero was often understood as “foot soldiers” who are sent to war without proper equipment (or even with no equipment), sufficient information, and adequate human forces and physical resources, and even without adequate support and compensation [43,44]. Rather, nurses are looking for recognition of their specific work and improvement of their working conditions [38,44,45,46].

Having worked in challenging conditions during the pandemic has prompted the participating nurses to consider that caring for patients in this context has been a significant learning experience for them. They report this experience has contributed to their acknowledgment of the importance of their professional roles, and of the personal and professional growth they had been developing. Having seen such difficult situations, they feel they can care for a variety of patients after the pandemic, and they feel that this experience will improve their development as nursing professionals. These results follow the line of studies carried out in other countries pointing to that the experience of having worked as a nurse during the COVID-19 pandemic helped the nurses discover their potential and realize the importance of their professional responsibilities [39]. Others inform that working during this pandemic accelerated the advancement of their nursing practice [47]. This pandemic helped nurses safely adapt to rapid decision making in situations of constant change [48,49]. Nursing care during the first two waves of the pandemic has focused on providing safe and efficient care. Most importantly, there has been a focus on providing patients and their families with care that is continuously based on current scientific evidence, and the dispersing of this information to them [48].

## 5. Limitations and Strengths

This research sample may be small in comparison to other works carried out. However, this study has focused on a specific research population of nurses who have been working in healthcare system hospitals of the provinces included in the study during the COVID-19 outbreak. The sample has served to saturate the information.

The participants of this study were nurses proficient in the hospital and intensive care units of Extremadura and Madrid. However, the experiences of caring for COVID-19 patients may differ according to the size of the hospital, work unit, region, and gender. Due to this, future studies should explore the experiences of sample participants with diverse characteristics.

## 6. Conclusions

This study reveals important aspects of the experiences of nurses in crisis situations such as the COVID-19 pandemic. Some recurring themes were the perspectives of the nurses regarding their professionalism, the fear of working in difficult situations dealing with uncertainty, and social reactions to what may happen.

The social and professional development of nurses is important. If nurses feel that they are quality professionals, this will enable them to protect their psychosocial health and increase professional commitment toward their patients in difficult situations such as pandemics.

The results of this study may serve as a guide for better understanding the problems and needs of nurses as healthcare providers. This may help administrators in the generation of solutions for the establishment of a safe and reliable work environment, which will in turn promote a healthcare system that can efficiently respond to future catastrophes.

According to the results of this study, institutional support for healthcare professionals in future pandemics should be improved, providing them with psychological support, material resources, and training to face the challenges.

## Figures and Tables

**Table 1 healthcare-11-02659-t001:** Main characteristics of participants.

Participants’ Demographics Characteristics (*n* = 14)		*n*
Age	<30 years old	2
	30–39 years old	8
	40–49 years old	2
	>50 years old	2
Gender	Male	3
	Female	11
Highest academic qualification	Bachelor’s Degree	6
	Specialist	1
	Master	6
	Ph.D.	1
Type of work	Temporary employment	9
	Fixed-term contract	3
	Permanent contract	2
Type of unit	Intensive Care	6
	Emergency	2
	Medical unit	6
Change of unit during COVID-19 crisis	Yes	3
	No	11
Years in practice	0–4 years	2
	5–10 years	5
	11–15 years	3
	More than 25 years	4

**Table 2 healthcare-11-02659-t002:** Interview topic script.

- Perceptions of nurses who worked on the frontline caring for people with COVID-19 on their professional commitment.
- Perceptions of nurses who worked on the frontline caring for people with COVID-19 on professional development during the first and second waves of COVID-19.

**Table 3 healthcare-11-02659-t003:** Categories, codes, and verbalizations of theme 1: professional engagement of nurses during the pandemic.

	Theme 1: Professional Engagement of Nurses during the Pandemic	
Categories	Codes	Verbatims
Commitment to continuing care	Not being able to stay at home	“I was afraid, but I didn’t think about leaving because I wanted to do everything I could. As much as I could, to help and take care of the patients… And I talked to my colleagues who stayed, and they said the same thing” (P. 08H32).
	Working more shifts	“I volunteered for three shifts and worked more than my allotted hours. Sometimes we left a bit later, mainly to give a hand to our colleagues” (P. 03M38).
	Caring by overcoming moments of weakness	“When I had a down period, I thought <<I have to keep going>>. We couldn’t afford to give up, so I have always been strong. This is not going to be enough for us and we must keep fighting and helping people so that they can move forward and be saved/healed” (P. 06M52).
	Nurses caring in the front line while doctors took a step backwards	“Those who entered the room were always nurses and assistants. The doctor was very afraid. The doctors trusted us, what we told them and what we didn’t, we valued… It is true that they gave us a lot of freedom, but the problem was that you were often helpless. It was a totally self-interested trust, and so they avoided going in to see the patient. They avoided contact with COVID-19, basically” (P. 08H32).
Living a historic moment	Pride in working together to alleviate the pandemic	“I think it was a little bit my personality, I am always willing to help with everything that has to do with my profession and a little bit the restlessness of seeing it from home, on TV and everything that was happening all over Spain. Motivated by this, by wanting to help, I joined the job as a nurse... In this dramatic situation I was happy to do my bit… it is true that I was afraid of getting infected, but I lived with it day by day and even risked my life to help solve it” (P. 01H24).
To be a psychological support for patients	Replacing family members	“For the patients we were someone else in their family, we became their family, the emotional bond was very strong. At the beginning they couldn’t talk to their family because the video calls took a long time, and they were in a room alone. They felt helpless and alone because they had no one next to them. When we came in to give them medication, to cure them or something, you tried to be their family for a while, to give them twice as much affection... you always give affection to the patients, but this time you had to do a lot more, the fact that they were dying alone was hard for us” (P. 06M52).
	Outside working hours	“Many times, when I finished my working day, I would go into the rooms to give them love and tell them <<come on, you can get out of this>>, <<in no time at all you will have your relatives, you will be with them>>. I have talked a lot with the patients, mainly about how they can get out of this, that they had us there to support them in everything, even if we couldn’t... even if their relatives weren’t there, they could rely on us psychologically and that they would get through it”(P. 06M52).

**Table 4 healthcare-11-02659-t004:** Categories, codes, and verbalizations of theme 2: impact of the pandemic on professional development.

	Theme 2: Impact of the Pandemic on Professional Development		
Categories	Codes		Verbatims
Change in the model of care and the meaning of care	Model focused on disease and population healthcare	No time to get to know patients	“What I like about my profession is that you relate to people, you know their names, you know the main caregiver. For me, the relationships I have with patients and careers are important, and that has not existed this time” (P. 09M52).
		Neglecting essential aspects of care	“What happened is that at the beginning we could not give them the level of care that you would have given to another type of patient without COVID-19, because you thought that it was so contagious, that everything was so dangerous. In fact, on one occasion we were called for an emergency, and you couldn’t enter the room until you were dressed in PPE, and you entered the room as little as possible” (P. 07M32).
		Protective measures hinder caring gestures	“A nurse always tries to be close to the patient, to deal with them hand in hand and of course with all those PIDs… You put yourself in that situation and you thought that the patient saw you as someone strange, someone who came with a diving mask to see them, to do anything, to take an IV, to give them a pill... the beginnings were hard” (P. 08H32).
		Frustration with the change of model	“I think that if there had been more information, the care would have been better, and it wouldn’t have generated so much frustration because in the end I think that it is a defense mechanism to work as an additional isolation” (P. 07M32).
	Humanizing care despite the situation	Providing comfort, support, and closeness	“Many times when I finished my working day I would go into the rooms to give them love, to tell them <<come on, you can get out of this>>, <<in no time at all you will have your relatives, you will be with them>>. I have talked a lot with the patients, mainly about how they can get out of this, that they had us there to support them in everything, even if we couldn’t… even if their relatives weren’t there, they could rely on us psychologically and that they would get through it”(P. 06M52).
Changes in work organization and structures	Changes in the nursing team (nurse and auxiliary)	Increased cohesion, solidarity, mutual help, and support among peers	“Before, there have always been groups at work, but when all this started, the truth is that we have all gone together, all as one and we have lent each other a hand, <<now I go into a box and then you go in>>. With this situation we have worked more as a team, we have helped each other, people have been very involved. The positive thing I take away with me is that all the colleagues have been a team, and have been always there, both when you were in hospital and at home” (P. 03M38).
		Teamwork	“With the pandemic we got to know again how nurses and assistants used to work together. Before, we worked each on our own. Not now, now you had to work together and whether you wanted to or not, there were more links, you helped the auxiliary to do hygiene… The truth is that we were much more of a family, and that brought us together a lot. The point was to help and get the job done, if you had to feed a dependent patient, you gave it to them. What was intended was that whoever came into the room should do all the care, and that is why we nurses have been the ones who have carried it out. That part was hard, but it was rewarding because we worked a lot as a team. Teamwork came back” (P. 05H30).
		Emotional support and comfort among peers	“Sitting down, even if it’s just for a while with a mask and two metres away, commenting, talking, de-stressing with colleagues, which is the only way to de-stress. Because it is true that some colleagues said <<I live with my family. I get there and I sleep in a different room from my husband, I’m afraid of infecting my son...>>, and the truth is that that was the moment when you could laugh a little, you could sit down and relax. After two or three hours with an EPI on, sweating, the truth is that it was gratifying to sit at that moment and disconnect a little with your colleagues who were the only friends you had, you know? I was happy to go to work because I couldn’t go out, you couldn’t relate to anyone. The only way to socialise and see other faces was at work, and if you could join in and talk to them and have a laugh that was our most rewarding part of the day” (P. 05H30).
	Changes in relations with doctors	Doctors in third line	“Before, the doctors stayed on the ward, they had their office, they had meetings with you, now they hardly ever went on the ward and they disappeared very quickly. It was the nurses and the assistants who were on the ward, and who managed everything. I think that because of the risk of infection and the fear that there was…, the atmosphere was already dirty, there were microorganisms in the environment. Before, they stayed with us, they stayed to write where we wrote, but not anymore. At that point they disappeared, and we had to contact them all day long by phone…” (P. 05H30).
		No teamwork	“The doctors went more at their own pace. Now we were servants of the doctors. When they came to visit, which was not every day… they were five-second visits and we had to be opening and closing the door for the doctor, assisting the doctor every time he had to auscultate a patient… in other words, we had to clean everything, leave it on the clean tray, he takes it and goes to auscultate another patient. And we had to write everything down for them when we already had our workload, and what we also complained about was that they could also work together, that one could open the door for the other” (P. 05H30).
		Doctors avoided contagious situations	“I am tremendously disappointed because we have experienced the most absolute loneliness. The ICU has moved forward thanks to the nursing team because the doctors didn’t go in, they were so scared that they didn’t go in. They even sent the resident, he was the only one who dressed up, went in with a tablet, took photos of the graphs and blood gas readings and left, that was it” (P. 07M32).
		Referrals to nurses for actions outside of their scope of competence	“We (nursing staff) saved the situation perfectly. If the doctor trusts you and he knows that you have entered the rooms and knows that you value the patient well and trusts you, you are saving him a lot of time. Maybe the internists would go in if they had to, otherwise you would take a lot of work away from them and they would get through the floor visit in half an hour” (P. 04M46).
Change in the social vision of nursing	Videos and dances on television and social media	Unrealistic image and missed opportunity to make nursing work more visible	“I am very much a dancer. We had the opportunity to have been seen as nurses a lot more and a lot better than we have been seen. But I don’t think it’s not just our fault or whether we dance or not, but people don’t know what a nurse does. You ask and they say <<that’s the one who changes nappies>>, <<that’s the one who orders this medication>>, they have no idea what a hospital nurse, a health centre nurse, school nurses, nurses in management positions and in universities, I don’t even know. I think we have missed an opportunity to empower ourselves and say <<hey, nurses do more than just dances and more than just give IVs>>. It’s true that everything is taken out of context, there are days when you don’t stop, and there are days when, if you have ten minutes at night, the patients are sleeping, and you have a dance and maybe that helps you get through the rest of the night. And thank goodness. People used to say <<no, people laugh>>, no, many patients have died with the coronavirus and before, that is, many patients have died and the fact that I laugh with my colleagues doesn’t mean I don’t feel it, it means that I cry for every patient who has died… Some days you cry more, others you don’t cry at all, others you forget them more quickly, and others you will always remember them, but just because you do a TikTok dance with your colleagues doesn’t mean that you are a nurse at all” (P. 10M30).
	The need for social and institutional support		“Nursing is a very vocational profession, but I think that health and nursing as such should be valued more. It is not because it is a vocation that one lives as a vocation, it is necessary that there be recognition both socially and the improvement of working conditions” (P. 05H30).
	Health workers as heroes		“The fact that we are considered heroes, I think, made the bar for all professionals too high. Because we are not heroes, we are people who have to go to work with our fears and uncertainties, and maybe the fact that you are considered like that can make people even more demanding. I know of people who will have psychological consequences and who will have to be treated” (P. 02M45).
	From heroes to villains		“How can you stop applauding the day the state of emergency is lifted? During the state of emergency you were a hero; when the state of emergency is lifted, you are no longer a hero, right? I feel that the health workers are heroes every day because they are fighting for you, but you value them one day, and the next day they are no longer heroes. So, I am very disappointed by society in that sense. I love my profession, but I am disappointed that it is not valued in that sense of saying <<today you are on top and tomorrow you are the one I insult, the one I attack… >>” (P. 5H30).
Improvements in professional development after the pandemic	Improving psychological skills	Strength, self-esteem, adaptability, and resilience	“I am stronger now, not more professional. Stronger, yes. With greater self-esteem, even capable of overcoming limits or reaching certain limits that you thought you were not going to be able to withstand and you have withstood them” (P. 07M32).
		Self-concept and flexibility	“I have realized that nurses can do anything. I mean, if you thought you couldn’t do certain things, in the end you get the job done. Nurses are a bit like that because we are nurses. Because in the end, every time you are sent to a new place, to a different place, you have a hard time, but in the end you get the job. I see this quality as inherent to the profession. <<We are nurses>>. Whatever they give us, in the end we do it, in the end you can cope with that and more” (P. 10M30).
	Teamwork skills	Connection, mutual support, solidarity, and cooperation	“As time went by, we realized that we had to change the way we worked and focus on the auxiliary-nurse team, all as one. And the truth is that we did it very well, and I am very proud of that whole period” (P. 08H32).
	Self-directed learning and lifelong learning	Self-training outside their specialty	“It has forced me to be trained in things outside my field of work, such as the use of respirators, parameters…” (P. 07M32).
		Improving the search for scientific evidence	“We looked for protocols on the internet because at that time nobody told us anything, nobody trained us in anything, neither in the placement of the EPI, nor in how to receive patients, nor protocols about anything… We used to search the internet to find out how to manage these patients, what they needed” (P. 07M32).
		Improved transmission of scientific evidence among peers	“We were looking for experiences of colleagues in Madrid, how they were coping, what was working for them, what was going wrong…” (P. 07M32).
		Self-training outside working hours	“My days off were spent watching videos on the internet about ventilators, to handle the ventilators to the last detail. I watched a thousand protocols, I looked at a thousand things, I trained myself daily. I had to get out of the ICU and look for information, because since I was isolated, I couldn’t disconnect. 24 h a day thinking about <<what can I do to improve this?>>” (P. 07M32).
	Improvement of professional self-concept	Opportunity to develop as a nurse	“I have been fortunate to have worked in a COVID-19 critical unit. For me it has been fortunate. It has reaffirmed me professionally. I have felt fully developed and like I have done everything in my power. I was there at that time, and I consider it lucky to have been able to experience this so that I could get to know it” (P. 07M32).

## Data Availability

The data presented in this study are available on request from the corresponding author.

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
