# Peer review of "Nurses’ Perceptions Regarding Their Professional Commitment and Development during the COVID-19 Pandemic"

_healthcare, 2023, doi:10.3390/healthcare11192659_

Round 1

Reviewer 1 Report

Introduction

While the introduction presents statistics about the number of cases and the impact on healthcare professionals, there is a notable absence of references after several factual statements. For example, lines 59-61 mention the fears and changes in protocols faced by healthcare professionals, yet no citation is provided. Ensuring every fact or claim is well-referenced will strengthen the manuscript's credibility.

The use of phrases like "It is known that…" (lines 52, 70) comes across as a bit colloquial and might benefit from a more formal rephrasing. Additionally, providing specific figures or studies alongside such statements would substantiate the claim.

The piece alludes to the challenges brought about by changes in protocol and lack of PPE (lines 61-64). It might be beneficial to delve deeper into how these challenges uniquely impacted nurses compared to other healthcare professionals.

Towards the end of the introduction, the mention of protective factors (lines 83-86) provides a promising segue to the objective of the study. It might be enriching to provide more details about these protective factors, even if briefly, before delving into the study's objective.

The manuscript suffers from erratic referencing. For instance, lines 59-61, which detail the fears and challenges faced by healthcare professionals, lack appropriate citations. It's imperative that all claims, especially in an academic context, are backed by verifiable sources. This oversight undermines the paper's reliability.

The recurrent use of "It is known that…" (as seen in lines 52 and 70) is overly casual for an academic document and lacks precision. Instead of assuming what's "known," the manuscript should provide concrete evidence or data to corroborate its statements.

The introduction makes broad claims about the challenges faced by nurses due to changes in protocols and shortages of PPE (lines 61-64). However, these statements are superficial and lack the nuance one would expect in a scholarly article. How did these issues uniquely affect nurses, as opposed to other healthcare personnel? Elaboration is crucial here.

While the research objective in lines 88-89 is explicit, it would benefit from a more detailed outline of the methodology and expected outcomes. As it stands, the objective seems detached from the preceding content, leaving the reader unclear about the paper's direction.

The manuscript repeatedly emphasizes the challenges faced by healthcare professionals, particularly nurses. While this is central to the topic, some statements seem redundant. Streamlining these would make for a more concise and impactful introduction.

The transition between the negative impacts of the pandemic and the protective factors (lines 83-86) is abrupt. The introduction would benefit from a more logical flow, perhaps by providing a clearer contrast between the challenges faced and the resilience shown by nurses.

The mention of "resilience and Spanish society’s support" (lines 85-86) as mitigating factors for the challenges faced by nurses appears to be based more on opinion than factual evidence. The paper should avoid making claims without providing robust empirical backing.

Finally, the objective is well-stated, but it might be clearer if the authors specify the methodology briefly, allowing readers to anticipate the direction the manuscript will take.

Methods:

The choice to utilize Giorgi’s descriptive phenomenological focus ([19,20] on lines 95-97) needs further justification. Why was this particular method preferred over other qualitative approaches?

The phrase "in their own words" is repetitively mentioned on lines 99 and 110. This redundancy should be addressed to ensure conciseness.

The selection of Extremadura and Madrid (lines 106-108) is briefly justified by their high prevalence of cases, but a more detailed justification, perhaps highlighting their unique characteristics or contrasts, could add depth to the understanding.

It's mentioned that data collection continued "until saturation of data" (lines 113-114). While this is standard for qualitative research, elaborating on the signs or indicators of this saturation may offer readers a clearer understanding of the research process.

The exclusion criteria that nurses who worked less than 2 months with COVID-19 patients (lines 125-126) seems arbitrary. How was this duration determined to be significant?

The manuscript mentions the principal investigator used an "outline of topics" (lines 129-131), but doesn't specify what these topics were. This information would be essential for other researchers seeking to replicate the study or understand its depth.

The observation that no significant differences were found between interviews conducted in person and via videoconference (lines 133-134) requires a more detailed explanation. On what criteria were these differences evaluated?

The steps of Giorgi’s phenomenological method are listed (lines 139-143) but not elaborated upon. It might benefit the reader to have a brief description of each step, particularly for those unfamiliar with this method.

The section on trustworthiness (lines 148-159) effectively maps out the criteria for rigor, but specific examples of how each criterion was met might strengthen the paper's claims to rigor and validity.

While the study claims to adhere to the Declaration of Helsinki (lines 161-162), it's crucial to also detail how participant welfare was protected, especially given the sensitive nature of discussing pandemic experiences.

Lines 166-167 mention that interviews were made anonymous before being analyzed. How was anonymity ensured? Were specific codes or pseudonyms assigned? Such details can bolster the study's ethical foundation.

The manuscript seems to skip some references, as noted from the sudden jump from [21] to [23] without a [22] (lines 102-108). This may indicate missing content or oversight in referencing.

Discussion

While the manuscript provides detailed findings, a clear elaboration on the methodology, particularly how participants were chosen and the methods of data collection and analysis, would strengthen the paper. Understanding these aspects would be crucial for readers to appreciate the depth and context of the findings.

Lines 364-367 suggest that nurses served as the primary source of psychological support for patients. However, line 367 points to another study emphasizing the need for professional mental health support. It might be beneficial to delve deeper into this contradiction. Does it suggest that while nurses played a pivotal emotional role, professional psychological intervention was still a missing component?

Lines 375-378 highlight the lack of institutional support and a stressful work environment. It would be beneficial to have more concrete examples or qualitative data to further emphasize these points and understand the specific kinds of institutional challenges nurses faced.

The discussion around lines 400-404 regarding nurses assuming the roles of other healthcare professionals is particularly intriguing. Some elaboration on why physicians were reluctant to fulfill their roles would provide a clearer picture of the healthcare dynamics during the pandemic.

Lines 419-422 provide an essential point on the social image of nurses and the potential negative impacts of being seen as "heroes". It would be enriching to provide more insights or quotes from participants on how this perception impacted them personally and professionally.

While the study does an excellent job at capturing the experiences and challenges, a section on recommendations for future pandemics or crises, based on the insights gained, would add considerable value.

Author Response

Dear Reviewer

Enclosed you will find a revision of our manuscript.We would like to thank you for allowing us to revise and improve our manuscript. We also thank the reviewers for their thoughtful and constructive comments. We have considered all the suggestions and have incorporated them into the revised and updated manuscript.

The article has been revised and modified to improve the understanding of the study. Changes to the original manuscript are made with Microsoft Word's track changes, and we believe our manuscript is stronger because of these modifications. We hope this now helps to improve the overall readability and quality of the manuscript.

Below is a detailed point-by-point response to the reviewers' comments.

Reviewer 2 Report

Abstract

1. Please verify the keywords for their alignment with mesh terms.

Introduction

1. What is the novelty of this research? Are you confident that there are no similar investigations conducted previously?

Materials and methods

1. Why did you decide that 2 months of experience are sufficient to include them in the interview?

2. What does it imply to have worked actively and non-actively during the pandemic?

3. Table 3 is quite extensive, spanning 5 pages. It might be more advantageous to place it in supplementary materials. Furthermore, you are describing categories and codes within the results section.

4. Could you please confirm whether all nurses worked both during the first and the second waves of the pandemic?

5. What was the procedure for recruiting respondents within the specified institutions? How were participants selected?

6. Where was the data collected?

7. Were there any respondents who declined to participate in the interviews? If so, what were the reasons for their refusal?

8. Was triangulation employed in the study?

9. Did the field notes record?

10. Was the interview audio-recorded? If yes, was consent obtained for audio and video recording?

11. How did you ascertain that there was no significant difference between video conference interviews and in-person interviews? (Lines 134-135)

12. Which statistical software did you utilize to calculate the mean interview duration? Why did you not specify this in the statistical analysis section?

13. Line 146. In sentence “During this phase, the Software Atlas-ti 8.0 was utilized as an aid” for which specific phase was the Atlas.ti software employed?

14. Who conducted the interviews? What experience or training did the researcher have? What were the researcher's credentials? Was the researcher male or female?

15. In the inclusion criteria, it is stated that nurses participating in the study must have a minimum work experience of 2 months. However, in Table 1, when describing work experience, the categorization starts as 0-4. Does this imply that you recruited nurses with 0 monthss of experience for the study?

16. What was the basis for your categorization of work experience using this particular method? Where is the category for 15 to 25 years of experience?

Conclusions

1. What specific recommendations or guidelines do you propose to assist nurses regarding this issue?

Author Response

(The authors gave the same response as above.)

Round 2

Reviewer 1 Report

The authors have met most of my concerns. I recommend the publication of this work. 

Reviewer 2 Report

No more suggestion.